# ANCHOR & TRANSFORM:
# LEARNING SPARSE REPRESENTATIONS OF DISCRETE OBJECTS

## ABSTRACT

Learning continuous representations of discrete objects such as text, users, and items lies at the heart of many applications including text and user modeling. Unfortunately, traditional methods that embed all objects do not scale to large vocabulary sizes and embedding dimensions. In this paper, we propose a general method, Anchor & Transform (ANT) that learns sparse representations of discrete objects by jointly learning a small set of *anchor embeddings* and a *sparse transformation* from anchor objects to all objects. ANT is scalable, flexible, end-to-end trainable, and allows the user to easily incorporate domain knowledge about object relationships (e.g. WordNet, co-occurrence, item clusters). ANT also recovers several task-specific baselines under certain structural assumptions on the anchors and transformation matrices. On text classification and language modeling benchmarks, ANT demonstrates stronger performance with fewer parameters as compared to existing vocabulary selection and embedding compression baselines.

## 1 INTRODUCTION

Learning continuous representations of discrete objects such as text, users, and items lies at the heart of many applications including natural language processing, user modeling, and recommendation systems. The standard way to learn these continuous representations involves: 1) defining the *vocabulary* $V = \{e_1, ..., e_{|V|}\}$ as the set of all objects, and 2) learning a $|V| \times d$ *embedding matrix* that defines a $d$ dimensional continuous representation for each object. However, when $|V|$ is large (e.g. million of words in language modeling or millions of users in user modeling), this embedding matrix does not scale elegantly and can constitute more than $90\%$ of all trainable parameters in the model. As a result, there has been large interest in learning *sparse representations* of these discrete objects rather than the full embedding matrix for cheaper storage as well as faster training and inference.

In the following we outline the desiderata of methods to learn sparse representations of discrete objects: 1) *General-purpose:* Good representation learning methods should be as flexible, modular, and as general as possible to be easily adapted for various applications. 2) *End-to-end trainable:* It is advantageous to learn neural representations in an end-to-end manner to directly optimize for the task at hand (Liu et al., 2018). Similarly, we would like to enforce sparsity in our representations via task-specific measures. 3) *Domain knowledge:* The user should be able to easily integrate domain knowledge about object relationships. Domain knowledge can include external sources such as WordNet (Miller, 1995), ConceptNet (Liu & Singh, 2004), and knowledge graphs (Pujara & Singh, 2018) for text modeling, as well as clusters of users and items for user modeling (Zhang et al., 2012).

As a step towards the aforementioned desiderata, we propose a general method to learn sparse representations of discrete objects. Our approach, depicted in Figure 1, consists of two steps:

1) ANCHOR: Learn embeddings of a small set of anchor objects that are representative of all objects (§3.1). The user retains flexibility in defining these anchors and we show strong performance across a suite of initialization strategies including random, frequency, and clustering-based methods.

2) TRANSFORM: Learn a sparse transformation from anchors to all objects (§3.2). Each of the non-anchors should be easily *induced* by some transformation from (a few) nearby anchors. Domain knowledge can be incorporated through transformations between related objects (§3.3). The anchor embeddings and the sparse transformation are trained end-to-end for task specific learning.

We call the resulting method *Anchor & Transform*, or ANT for short. ANT is scalable, flexible, end-to-end trainable, and allows the user to easily incorporate domain knowledge about object relationships. We also show that ANT recovers several task-specific baselines under certain structural assumptions on the anchors and transformation matrices (Appendix D). In practice, ANT is easy

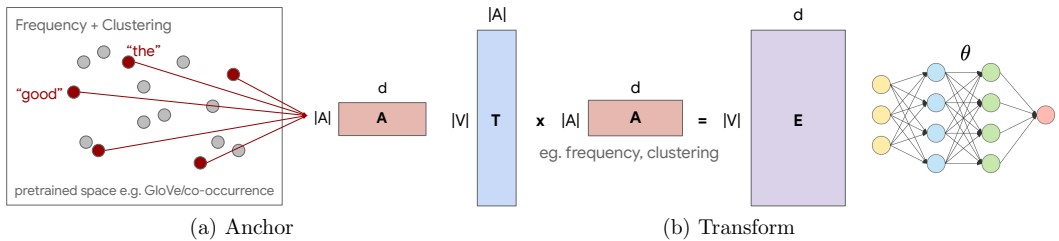

(a) Anchor                 (b) Transform

Figure 1: Anchor & Transform (ANT) is a general method to learn sparse representations of discrete objects consisting of two steps: 1) *Anchor:* Learn embeddings **A** of a small set of anchor objects $A = \{a_1, ..., a_{|A|}\}, |A| << |V|$ that are *representative* of all discrete objects (e.g. frequency and clustering based initialization methods). 2) *Transform:* Learn a sparse transformation **T** from the anchor embeddings to the full embedding matrix **E**. **A** and **T** are trained end-to-end for task specific representation learning. ANT is scalable, flexible, end-to-end trainable, and allows the user to easily incorporate domain knowledge about object interactions.

to incorporate into existing models by replacing the EMBEDDING layers in PyTorch or TensorFlow with our newly designed ANTEMBEDDING layers. We demonstrate the effectiveness of ANT on text classification and language modeling tasks (§4). ANT achieves better performance with fewer parameters as compared to existing vocabulary selection and embedding compression baselines. Code to replicate our experiments is publicly released at <anonymous>.

## 2 RELATED WORK

Related work on learning sparse representations of discrete structures largely fall into three categories:

**General purpose matrix compression techniques** have been used to reduce the embedding matrix **E**, such as low rank approximations (Acharya et al., 2018; Markovsky, 2011; Grachev et al., 2019), quantizing (Han et al., 2016), pruning (Wen et al., 2016; Dong et al., 2017; Anwar et al., 2015), or hashing (Guo et al., 2017; Chen et al., 2015; Qi et al., 2017). However, these methods do not allow the user to integrate domain knowledge about object relationships. Furthermore, their generality implies that they are complementary and can be combined with our proposed methods.

**Reducing representation size:** These methods reduce the dimension $d$ for different objects. For example, Chen et al. (2016a) divides the embedding into buckets which are assigned to objects in order of importance, Joglekar et al. (2019) learns $d$ by solving a discrete optimization problem with reinforcement learning (RL), and Baevski & Auli (2018) gradually reduces dimensions for rarer words. These methods typically resort to RL or are difficult to tune with too many hyperparameters. Each object is also modeled independently without information sharing between objects.

**Task specific methods** define sparse embeddings for specific applications. For example, learning representations for common words gives strong performance for language modeling (Luong et al., 2015; Chen et al., 2016b), and performing dropout-based vocabulary selection retains performance on text classification (Chen et al., 2019). Other methods aim to reconstruct pre-trained embeddings using codebook learning (Shu & Nakayama, 2018; Chen et al., 2018) or low rank tensors (Nam & Quoc, 2017; Sedov & Yang, 2018). However, these methods are not general and end-to-end trainable, therefore preventing us from learning representations for general tasks. For example, methods that only model a subset of objects cannot be used for retrieval tasks because it would imply never retrieving the dropped objects. Rare objects might be highly relevant to a few users so it might not be ideal to completely ignore them, but rather use only a few parameters to model them. Similarly, subword (Bojanowski et al., 2016) and wordpiece (Wu et al., 2016) embeddings, while useful for text, do not generalize to general objects and applications such as item and query retrieval.

The concept of using anchors to succinctly represent a large set of points on a manifold or space has been around for a while Mallat (1999); Aharon et al. (2006). However, to best of our knowledge, they are mostly limited to cases when a distance or similarity function or a proxy thereof like ranked lists are provided (Guo et al., 2017; Xu et al., 2011; Liang et al., 2018). The use of anchors simultaneously with a downstream learning task has been limited to well understood convex objectives with continuous inputs like in sparse Gaussian Process Regression (Snelson & Ghahramani, 2006), where the kernel acts as the distance function. It is not evident from literature how to apply these anchor tricks to arbitrary learning problem over discrete objects in a flexible manner, like in deep networks. Our main contribution is to demonstrate how these anchors and sparse transformations

Figure 2: Initialization strategies for anchor objects combining ideas from frequency and $k$-means++ clustering algorithms. Clustering initialization picks anchors to span the space of all objects.

can be trained jointly with neural models as a general input embedding layer, and how we can obtain better sparse representations using domain knowledge, e.g. if some notion of similarity is provided.

## 3  ANCHOR & TRANSFORM

We present a *general purpose* method that is *end-to-end trainable* and allows us to *integrate domain knowledge* about object relationships. We suppose we are presented with data $\mathbf{X} \in \mathbb{R}^{n \times |V|}, \mathbf{y} \in \mathbb{R}^{n \times c}$ drawn from some joint distribution $p(X, y)$ where the support of $X$ is over a discrete set $V = \{e_1, ..., e_{|V|}\}$. $n$ is the size of the training set, $V$ is typically known as the vocabulary, and $|V|$ the vocabulary size. The entries in $\mathbf{X}$ are one-hot realizations of the discrete random variable $X$ while the entries in $\mathbf{y}$ can be either discrete (classification with $c$ classes) or continuous (regression with $c$ targets). The goal is to learn a $d$-dimensional continuous *representation* $\{\mathbf{e}_1, ..., \mathbf{e}_{|V|}\}$ for each discrete object by learning an embedding matrix $\mathbf{E} \in \mathbb{R}^{|V| \times d}$ where row $i$ is the representation $\mathbf{e}_i$ of object $i$. Learning this representation allows us to *embed* the discrete data into $\mathbf{H} = \mathbf{X}\mathbf{E}$ and define a neural model $f_\theta$ with parameters $\theta$ to predict $\hat{\mathbf{y}}$ given representations $\mathbf{H}$, i.e. $\hat{\mathbf{y}} = f_\theta(\mathbf{H}) = f_\theta(\mathbf{X}\mathbf{E})$.

Instead of learning the large embedding matrix $\mathbf{E}$ directly, ANT consists of two components:

1) ANCHOR: Learn embeddings $\mathbf{A} \in \mathbb{R}^{|A| \times d}$ of a small set of anchor objects $A = \{a_1, ..., a_{|A|}\}, |A| <<$ $|V|$ that are *representative* of all discrete objects. One is free to use different methods (e.g. random, frequency, or clustering-based initialization strategies) to initialize/select anchors based on the task.

2) TRANSFORM: Learn a sparse transformation $\mathbf{T}$ from $\mathbf{A}$ to the full embeddings $\mathbf{E}$. In other words, each of the non-anchor objects should be easily *induced* by some transformation from (a few) nearby anchor objects. To ensure sparsity, we want nnz$(\mathbf{T}) << |V| \times d$. Furthermore, domain knowledge can be incorporated by defining the structure of these transformations between related objects.

$\mathbf{A}$ and $\mathbf{T}$ are trained end-to-end for task specific representation learning. After training, we only store $|A| \times d + \text{nnz}(\mathbf{T}) << |V| \times d$ entries that define the complete embedding, which is much fewer than the $|V| \times d$ embedding traditionally used. We depict the overall algorithm in Figure 1. In the following subsections, we explain how these anchor embeddings (§3.1) and sparse transformations (§3.2) are learned, before describing how the user can incorporate domain knowledge into ANT (§3.3).

### 3.1  ANCHOR: LEARNING THE ANCHOR EMBEDDINGS $\mathbf{A}$

In the ANCHOR step, we would like to find a small set of anchor objects $A = \{a_1, ..., a_{|A|}\}, |A| << |V|$ that are representative of all discrete objects. We describe several methods to select these anchors[1].

**Frequency and TF-IDF:** For tasks where frequency or TF-IDF (Ramos, 1999) are useful for prediction, the objects can simply be sorted by frequency or TF-IDF and the most common objects selected as the anchor points. While this might make sense for tasks such as language modeling (Luong et al., 2015; Chen et al., 2016b), only choosing the most frequent objects might not cover the space of all objects. As a result, certain rare objects might not be well represented by the common anchors.

**Clustering initialization:** To ensure that all objects are close to some anchor in the representation space, we use $k$-means++ initialization (Arthur & Vassilvitskii, 2007). Given a predefined metric space $S$ consisting of $n$ points $\mathbf{x}_1, ..., \mathbf{x}_n \in \mathbb{R}^d$ with some distance function $d()$, $k$-means++ initialization is a randomized algorithm that picks cluster centers to span the entire space. In practice, $S$ is usually a feature space representative of the relationships between objects, such as the Glove (Pennington et al., 2014) or Word2vec (Mikolov et al., 2013) embedding space for words or a co-occurrence matrix (Haralick et al., 1973) for more general objects. Using $C$ to denote the set of cluster centers ($C = \{\}$ initially), the $k$-means++ initialization alternates between two steps: 1. $D(\mathbf{x}_i) = \min_{\mathbf{c} \in C} d(\mathbf{x}_i, \mathbf{c})$, and 2. add $\mathbf{x}_i$ to $C$ with probability $\propto D^2(\mathbf{x}_i)$. This clustering initial-

---

[1]Please refer to Appendix A for more strategies and a comparison of various initialization strategies.

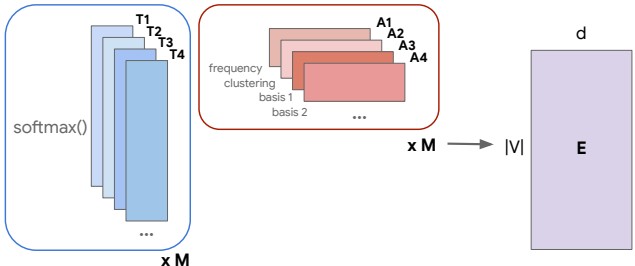

Figure 3: Generalized nonlinear mixture of anchors $\mathbf{A}_1, ..., \mathbf{A}_M$ and transformations $\mathbf{T}_1, ..., \mathbf{T}_M$, $\mathbf{E} = \sum_{m=1}^{M} \text{softmax}(\mathbf{T}_m)\mathbf{A}_m$ (softmax across rows of $\mathbf{T}_m$). Different sparse transformations can be learned for different initializations of anchor embeddings.

ization method can also be used to augment other strategies, such as initializing anchors using frequent objects followed by clustering iterations to complete the remaining anchors (Figure 2).

**Dynamic basis vectors:** Randomly initialize $\mathbf{A}$ to a set of basis vectors. This simple yet powerful method captures the general case where we have less knowledge about the objects. The random basis $\mathbf{A}$ can be viewed as a low-rank representation of the full embedding space $\mathbf{E}$.

**Mixture of anchors:** In general, different initialization strategies may bring about different advantages. For example, using a mixture of random basis vectors has been shown to help model multisense embeddings (Athiwaratkun et al., 2018; Nguyen et al., 2017). One can define a set of $M$ anchor embeddings $\mathbf{A}_1, ..., \mathbf{A}_M$ each initialized by different strategies and of possibly different sizes.

### 3.2 TRANSFORM: LEARNING THE SPARSE TRANSFORMATION $\mathbf{T}$

We now describe how to TRANSFORM the anchor embeddings $\mathbf{A}$ to the full embedding matrix $\mathbf{E}$, while enforcing sparsity in how these transformations are parametrized.

**Linear transformation:** We begin with a simple linear transformation $\mathbf{T} \in \mathbb{R}^{|V| \times |A|}$ that maps the anchor embeddings to the entire embedding matrix: $\mathbf{E} = \mathbf{TA}$. The final embedding $\mathbf{e}_i$ for object $i$ is given by a linear combination of anchor embeddings whose weights are given by $\mathbf{t}_i$ (row $i$ of $\mathbf{T}$).

**End-to-end training objective:** Given the embedding matrix $\mathbf{E}$ which has now been implicitly defined by $\mathbf{A}$ and $\mathbf{T}$, we add a model with parameters $\theta$ on top of the embedding matrix (e.g. fully connected layers or RNN/CNN/Transformer for sequential data) and define a loss function $\mathcal{L}$ which is a function of $\mathbf{A}$, $\mathbf{T}$, and $\theta$ (and $\mathbf{X}, \mathbf{y}$). Our end-to-end training objective is therefore

$$\mathbf{A}^*, \mathbf{T}^*, \theta^* = \arg\min_{\mathbf{A}, \mathbf{T}, \theta} \mathcal{L}(f(\mathbf{X}; \mathbf{A}, \mathbf{T}, \theta), \mathbf{y}), \tag{1}$$

which directly optimizes for $\mathbf{A}$ and $\mathbf{T}$ for task-specific representation learning.

**Making $\mathbf{T}$ sparse:** As it is, $\mathbf{T}$ is still high-dimensional with $|V| \times |A|$ parameters. To enforce sparsity, we add an $\ell_1$ penalty on transformation $\mathbf{T}$, which transforms our overall objective function into:

$$\mathbf{A}^*, \mathbf{T}^*, \theta^* = \arg\min_{\mathbf{A}, \mathbf{T}, \theta} \mathcal{L}(f(\mathbf{X}; \mathbf{A}, \mathbf{T}, \theta), \mathbf{y}) + \lambda \|\mathbf{T}\|_1, \tag{2}$$

where $\lambda$ is a regularization factor. $\|\mathbf{T}\|_1$ denotes sum of absolute values (not matrix nuclear norm). Most deep learning frameworks, would directly use subgradient descent to solve equation 2, but unfortunately such an approach will not yield sparsity. Instead, we perform optimization by proximal gradient descent (Parikh & Boyd, 2014) to ensure exact zero entries in $\mathbf{T}$:

$$\mathbf{A}^{t+1}, \mathbf{T}^{t+1}, \theta^{t+1} = \text{GRADIENT STEP}(\nabla\mathcal{L}(f(\mathbf{X}; \mathbf{A}^t, \mathbf{T}^t, \theta^t), \mathbf{y}), \eta), \tag{3}$$

$$\mathbf{T}^{t+1} = \text{PROX}_{\eta\lambda}(\mathbf{T}^{t+1}), \tag{4}$$

where $\eta$ is the learning rate, GRADIENT STEP is a gradient update rule (e.g. SGD (Lecun et al., 1998), ADAM (Kingma & Ba, 2014)), and PROX$_{\eta\lambda}$ is the soft-thresholding operator (Beck & Teboulle, 2009) with threshold $\eta\lambda$. This results in a transformation with few non-zero entries. Embedding $\mathbf{e}_i$ for object $i$ is now defined by a sparse combination of anchor embeddings with weights given by $\mathbf{t}_i$.

**Connections to sparse dictionary learning, sparse recovery, compressive sensing:** In Appendix B, we outline connections between ANT and sparse dictionary learning (Awasthi & Vijayaraghavan, 2018). By using results from sparse recovery (Candes & Tao, 2005; Candès, 2008), we can show that $\ell_1$-regularization (equation 2) results in provably sparse entries in $\mathbf{T}$.

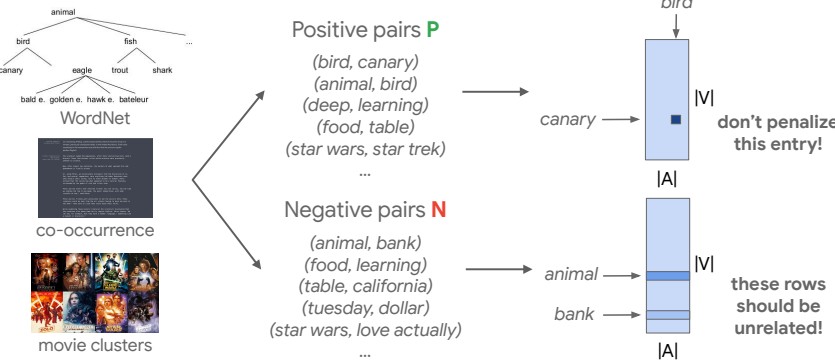

Figure 4: Incorporating domain knowledge via given relationship graphs (left) and extracting positive and pairs ($\mathbf{P}$) and negative pairs ($\mathbf{N}$). Transformations between negative (unrelated) pairs are sparsely penalized while those between positive (related) pairs are not. The linear combination coefficients $\mathbf{t}_u$ and $\mathbf{t}_v$ of negative pairs are also discouraged from sharing similar entries.

**Reducing redundancy in representations:** We additionally employ two methods to reduce redundancy in our sparse representations. Firstly, we perform orthogonal regularization of dynamic basis vectors $\mathbf{A}$ by adding the loss term $\mathcal{L}(\mathbf{A}) = \sum_{i \neq j} \left| \mathbf{a}_i^\top \mathbf{a}_j \right|$ to the loss function in equation 2. This ensures that different basis vectors $\mathbf{a}_i$ and $\mathbf{a}_j$ are orthogonal instead of being linear combinations of one another which would lead to redundancies across different learnt entries in $\mathbf{T}$. Secondly, we employ a non-negative constraint on $\mathbf{T}$ by passing it through a Relu activation: $\mathbf{T} = \text{RELU}(\mathbf{T})$, which ensures that positive and negative entries along the same row in $\mathbf{T}$ do not effectively cancel each other out.

**Nonlinear mixture of transformations:** In §3.1 we proposed to learn multiple sets of anchor embeddings $\mathbf{A}_1, ..., \mathbf{A}_M$. To truly exhibit the advantage of multiple anchors, we need to transform and combine them in a nonlinear fashion, e.g. $\mathbf{E} = \sum_{m=1}^{M} \text{softmax}(\mathbf{T}_m)\mathbf{A}_m$ (softmax over the rows of $\mathbf{T}_m$, Figure 3). There is a connection between nonlinear mixture of transformations with the multi-head attention mechanism used in the Transformer (Vaswani et al., 2017), where $\text{softmax}(\mathbf{T}_m)$ can be viewed as softmax-activated (sparse) attention weights and $\mathbf{A}_m$ the values to attend over.

### 3.3 INCORPORATING DOMAIN KNOWLEDGE

Another benefit of our approach is its generality in incorporating *domain knowledge* about object relationships. Suppose we are given some relationship graph $G = (V, E)$ where each object is represented by a vertex $v \in V$ and an edge $(u, v) \in E$ exists between objects $u$ and $v$ if they are related. Real-world instantiations of such a graph include 1) WordNet (Miller, 1995) or ConceptNet (Liu & Singh, 2004) where an edge exists between two words if they are related through semantic relations (e.g. synonym, antonym, hypernym, hyponym), 2) word co-occurrence matrices (Haralick et al., 1973) where an edge exists between two words if they frequently co-occur, and 3) Movie Clustering datasets (Leskovec & Krevl, 2014) where edges exist between two movies if they belong in the same cluster. From the relevant relationship graphs, we extract positive pairs $P = \{(u, v) : (u, v) \in E\}$ (which is the set of all related object pairs) and the unrelated negative pairs $N = \{(u, v) : (u, v) \notin E\}$. We incorporate the domain information from positive and negative pairs as follows:

**Positive pairs:** To incorporate a positive pair $(u, v)$, we *do not* enforce sparsity on the entries in the transformation matrix $\mathbf{T}$ which correspond to row $u$ and column $v$ (and vice versa). This allows ANT to freely learn the transformation between related objects $u$ and $v$ without being penalized for sparsity. On the other hand, transformations between negative pairs will be sparsely penalized. In other words, we element-wise multiply $\mathbf{T}$ with a *domain sparsity matrix* $\mathbf{S}(G)$ where $\mathbf{S}(G)_{u,v} = 0$ for $(u, v) \in P$ (entries not $\ell_1$-penalized) and $\mathbf{S}(G)_{u,v} = 1$ otherwise (entries are $\ell_1$-penalized).

$$\mathbf{A}^*, \mathbf{T}^*, \theta^* = \underset{\mathbf{A}, \mathbf{T}, \theta}{\arg\min} \, \mathcal{L}(f(\mathbf{X}; \mathbf{A}, \mathbf{T}, \theta), \mathbf{y}) + \lambda \|\mathbf{T} \odot \mathbf{S}(G)\|_1, \tag{5}$$

Since we perform proximal GD this is equivalent to only soft-thresholding the regularized entries between unrelated objects, i.e. $\mathbf{T} = \text{PROX}_{\eta\lambda}(\mathbf{T} \odot \mathbf{S}(G)) + \mathbf{T} \odot (\mathbf{1} - \mathbf{S}(G))$ (Kim & Xing, 2008).

**Negative pairs:** To incorporate information about negative pairs, we add an additional constraint that unrelated pairs should not share entries in their linear combination coefficients of the anchor embeddings. In other words, we add the loss term $\mathcal{L}(\mathbf{T}, N) = \sum_{(u,v) \in N} \left| \mathbf{t}_u \right|^\top \left| \mathbf{t}_v \right|$ to the loss function in equation 2. where each inner sum discourages $\mathbf{t}_u$ and $\mathbf{t}_v$ from sharing similar entries.

---

**Algorithm 1** Anchor & Transform algorithm for learning sparse representations of discrete objects.

---

**Ensure** **ANCHOR & TRANSFORM:**
1: Anchor: initialize anchor objects $A$ and their embeddings $\mathbf{A}_1, ..., \mathbf{A}_M$.
2: Transform: initialize linear transformations $\mathbf{T}_1, ..., \mathbf{T}_M$ as *sparse matrices*.
3: Optionally + domain info: initialize domain sparsity matrix $\mathbf{S}(G)$ as a *sparse matrix*.
4: **for** $(\mathbf{X}, \mathbf{y})$ in each batch **do**
5:     Construct embedding matrix $\mathbf{E} = \sum_{m=1}^{M} \text{softmax}(\mathbf{T}_m)\mathbf{A}_m$.
6:     Embed input data tokens $\mathbf{X}$ with embeddings $\mathbf{E}$.
7:     Compute predictions $\hat{\mathbf{y}} = f_\theta(\mathbf{X}, \mathbf{A}, \mathbf{T})$.
8:     Compute loss $\mathcal{L} = \mathcal{L}(f(\mathbf{X}; \mathbf{A}, \mathbf{T}, \theta), \mathbf{y}) + \mathcal{L}(\mathbf{A}) + \mathcal{L}(\mathbf{T}, N)$ (optionally + domain info).
9:     $\mathbf{A}, \mathbf{T}, \theta = \text{GRADIENT STEP} (\nabla \mathcal{L}, \eta)$.
10:     $\mathbf{T} = \text{RELU}(\text{PROX}_{\eta\lambda}(\mathbf{T} \odot \mathbf{S}(G)) + \mathbf{T} \odot (\mathbf{1} - \mathbf{S}(G)))$.
11: **end for**
12: **return** anchor embeddings $\mathbf{A}$ and sparse transformations $\text{nnz}(\mathbf{T})$.

---

### 3.4 EFFICIENT TRAINING AND INFERENCE

The overall algorithm is given in Algorithm 1. We take advantage of *sparse matrices* during training and inference. Specifically, $\mathbf{T}$ is implemented as a sparse matrix by only storing its non-zero entries and indices. From our experiments, we observe that $\text{nnz}(\mathbf{T}) << |V| \times d$ which makes storage of $\mathbf{T}$ extremely efficient as compared to traditional approaches of computing and storing the entire $|V| \times d$ embedding matrix. We also provide some implementation tips to further speedup training in Appendix C and ways to incorporate ANT with existing speedup techniques like softmax sampling (Bengio & Senecal, 2008; Mikolov et al., 2013) or noise-contrastive estimation (Gutmann & Hyvrinen, 2010; Mnih & Teh, 2012). After training, we only store $|A| \times d + \text{nnz}(\mathbf{T}) << |V| \times d$ entries that completely define the complete embedding matrix, thereby using fewer parameters than the traditional $|V| \times d$ embedding matrix. Furthermore, we emphasize that *general purpose matrix compression techniques* such as hashing (Qi et al., 2017), pruning (Dong et al., 2017), and quantizing (Han et al., 2016) are compatible with our method: the resulting matrices $\mathbf{A}$ and $\text{nnz}(\mathbf{T})$ can be further compressed and stored.

### 3.5 GENERALITY OF ANT

In Appendix D, we show that under certain structural assumptions on the anchors and transformation matrices, ANT reduces to several established task-specific methods for learning sparse representations: 1) Frequency (Luong et al., 2015; Chen et al., 2016b), TF-IDF (Ramos, 1999), Group Lasso (Wen et al., 2016), and variational dropout (Chen et al., 2019) based vocabulary selection, 2) Low-rank factorization (Acharya et al., 2018; Grachev et al., 2019), and 3) Compositional code learning (Shu & Nakayama, 2018; Chen et al., 2018). Hence, ANT is general and unifies some of the work on sparse representation learning done independently in different areas.

## 4 EXPERIMENTS

To evaluate the effectiveness of ANT in learning sparse representations of discrete objects, we evaluate performance on text classification and language modeling tasks.

### 4.1 TEXT CLASSIFICATION

**Setup:** We follow the setting in Chen et al. (2019) with four datasets: AG-News ($|V| = 62$K) (Zhang et al., 2015), DBPedia ($|V| = 563$K) (Lehmann et al., 2015), Sogou-News ($|V| = 254$K) (Zhang et al., 2015), and Yelp-review ($|V| = 253$K) (Zhang et al., 2015). We use a CNN for classification (Kim, 2014). ANT is used to replace the input embedding and domain knowledge is derived from WordNet and co-occurrence in the training set. We record test accuracy and number of parameters used in the embedding only (excluding CNN). For ANT, num params is computed as $|A| \times d + \text{nnz}(\mathbf{T})$.

**Baselines:** In addition to the **CNN**, we compare to the following compression approaches. Vocabulary selection: 1) **Frequency** where only embeddings for most frequent words are learnt (Luong et al., 2015; Chen et al., 2016b), 2) **TF-IDF** which only learns embeddings for words with high TF-IDF score (Ramos, 1999), 3) **GL** (group lasso) which aims to find underlying sparse structures in the embedding matrix via row-wise $\ell_2$ regularization (Park et al., 2016; Liu et al., 2015; Wen et al., 2016), 4) **VVD** (variational vocabulary dropout) which performs variational dropout for vocabulary selection (Chen et al., 2019), 5) **SparseVD** (sparse variational dropout) which performs variational

Table 1: Text classification results on AG-News. Domain knowledge is derived from WordNet and co-occurrence statistics. Our approach with different initializations and domain knowledge achieves within $0.5\%$ accuracy with $40\times$ fewer parameters, outperforming the published baselines. Init: initialization method, Acc: accuracy, # Emb: # (non-zero) embedding parameters.

| Method | $\|A\|$ | Init $A$ | Sparse $\mathbf{T}$ | RELU($\mathbf{T}$) | Domain | Acc (%) | # Emb (M) |
|---|---|---|---|---|---|---|---|
| CNN (Zhang et al., 2015) | 61,673 | All | ✗ | ✗ | ✗ | 91.6 | 15.87 |
| Frequency (Chen et al., 2019) | 5,000 | Frequency | ✗ | ✗ | ✗ | 91.0 | 1.28 |
| TF-IDF (Chen et al., 2019) | 5,000 | TF-IDF | ✗ | ✗ | ✗ | 91.0 | 1.28 |
| GL (Chen et al., 2019) | 4,000 | Group lasso | ✗ | ✗ | ✗ | 91.0 | 1.02 |
| VVD (Chen et al., 2019) | 3,000 | Var dropout | ✗ | ✗ | ✗ | 91.0 | 0.77 |
| SparseVD (Chirkova et al., 2018) | 5,700 | Mult weights | ✗ | ✗ | ✗ | 88.8 | 1.72 |
| SparseVD-Voc (Chirkova et al., 2018) | 2,400 | Mult weights | ✗ | ✗ | ✗ | 89.2 | 0.73 |
| Sparse Code (Chen et al., 2016b) | 100 | Frequency | ✓ | ✗ | ✗ | 89.5 | 2.03 |
| Anchor & Transform | 50 | Frequency | ✓ | ✓ | ✗ | 89.5 | 1.01 |
| | 10 | Frequency | ✓ | ✓ | ✓ | **91.0** | **0.40** |
| | 10 | Dynamic | ✓ | ✓ | ✓ | 90.5 | 0.40 |
| | 5 | Dynamic mixture | ✓ | ✓ | ✓ | 90.5 | 0.70 |

dropout on all parameters (Chirkova et al., 2018), and 6) **SparseVD-Voc** which additionally uses multiplicative weights for vocabulary sparsification (Chirkova et al., 2018). 7) We also compare to a **Sparse Code** model that learns a sparse code to reconstruct pretrained word representations (Chen et al., 2016b). All CNN architectures are the same throughout these baselines, details in Appendix E.1.

**Results** on AG-News are presented in Table 1 and we provide results for other text classification datasets in Appendix F. We observe that passing $\mathbf{T}$ through a RELU function is important in reducing redundancy in the linear combination entries. Domain knowledge from WordNet and co-occurrence statistics also succeeded in reducing the total (non-zero) embedding parameters to $0.40$M, a compression of $40\times$ the baseline and outperforming the existing approaches. Using a mixture of anchors and transformations also achieves stronger performance than the baselines using $5$ anchors per mixture, although the larger number of transformations leads to an increase in parameters.

## 4.2 LANGUAGE MODELING

**Setup:** We perform experiments on word-level Penn Treebank (PTB) ($|V| = 10$K) (Marcus et al., 1993) and WikiText-103 ($|V| = 267$K) (Merity et al., 2016). We experiment with both vanilla LSTMs (Hochreiter & Schmidhuber, 1997) and AWD-LSTM (Merity et al., 2017) for language modeling. We use ANT to replace the input embedding and the output embedding is tied to the input. Domain knowledge is derived from both WordNet and co-occurrence statistics on the training set. We record the test perplexity and the number of (non-zero) embedding parameters.

**Baselines:** We compare to **SparseVD** and **SparseVD-Voc**, as well as low-rank (**LR**) and tensor-train (**TT**) model compression techniques (Grachev et al., 2019). Since Grachev et al. (2019) train LSTM models with different embedding sizes (200 and 256), we experiment with both as well (details in Appendix E.2). Note that the application of variational vocabulary selection to language modeling with tied weights is non-trivial since one is unable to predict next words when words are dynamically dropped out. We also compare against methods that compress the trained embedding matrix as a *post-processing* step before evaluating using the compressed embedding matrix: **Post-SH** (post-processing using sparse hashing) (Guo et al., 2017) and **Post-SH+k-SVD** (improving sparse representation using k-SVD) (Guo et al., 2017; Awasthi & Vijayaraghavan, 2018) which additionally uses k-SVD (Aharon et al., 2006) to solve for a sparse representation of the embedding matrix, instead of adhoc-projection in Guo et al. (2017, eq 8-9). Comparing to these post-processing methods will demonstrate that end-to-end joint training of sparse embedding matrices is beneficial over post-processing compression. We also note that while these post-processing methods reduce the number of (non-zero) parameters required for storage and evaluation, the full embedding matrix still needs to be learned during training.

**Results:** On PTB (Table 2), we improve the perplexity and compression as compared to both previously proposed methods that use variational dropout and matrix compression techniques. We observe that sparsity is important: baseline methods that only perform lower-rank compression with dense factors (e.g. LR LSTM) tend to suffer in performance while using many parameters, while ANT retains performance with much better compression. ANT also outperforms post-processing methods (Post-SH and Post-SH+k-SVD), we hypothesize this is because these post-processing accumulate errors in both language modeling as well as embedding reconstruction. Using an anchor size of

Table 2: Language modeling using LSTM (top) and AWD-LSTM (bottom) on PTB. We outperform vocab selection and compression baselines. 200/256 is embedding dimension. Incorporating domain knowledge further reduces params. Ppl: perplexity, # Emb: # (non-zero) embedding parameters.

| Method | $|A|$ | Init $A$ | Sparse $\mathbf{T}$ | RELU($\mathbf{T}$) | Domain | Ppl | # Emb (M) |
|---|---|---|---|---|---|---|---|
| LSTM 200 (Grachev et al., 2019) | 10,000 | All | ✗ | ✗ | ✗ | 77.1 | 2.00 |
| LSTM 256 (Chirkova et al., 2018) | 10,000 | All | ✗ | ✗ | ✗ | 70.3 | 2.56 |
| LR 200 (Grachev et al., 2019) | 10,000 | All | ✗ | ✗ | ✗ | 112.1 | 1.26 |
| TT 200 (Grachev et al., 2019) | 10,000 | All | ✗ | ✗ | ✗ | 116.6 | 1.16 |
| SparseVD 256 (Chirkova et al., 2018) | 9,985 | Mult weights | ✗ | ✗ | ✗ | 109.2 | 1.34 |
| SparseVD-Voc 256 (Chirkova et al., 2018) | 4,353 | Mult weights | ✗ | ✗ | ✗ | 120.2 | 0.52 |
| Anchor & Transform 200 | 2,000 | Dynamic | ✓ | ✓ | ✗ | **77.7** | 0.65 |
| | 1,000 | Dynamic | ✓ | ✓ | ✗ | 79.4 | 0.41 |
| | 500 | Dynamic | ✓ | ✓ | ✗ | 84.5 | 0.27 |
| | 100 | Dynamic | ✓ | ✓ | ✗ | 106.6 | **0.05** |
| Anchor & Transform 256 | 2,000 | Dynamic | ✓ | ✓ | ✗ | **71.5** | 0.78 |
| | 1,000 | Dynamic | ✓ | ✓ | ✗ | 73.1 | 0.49 |
| | 500 | Dynamic | ✓ | ✓ | ✗ | 77.2 | 0.31 |
| | 100 | Dynamic | ✓ | ✓ | ✗ | 96.5 | **0.05** |

| Method | $|A|$ | Init $A$ | Sparse $\mathbf{T}$ | RELU($\mathbf{T}$) | Domain | Ppl | # Emb (M) |
|---|---|---|---|---|---|---|---|
| AWD-LSTM (Merity et al., 2017) | 10,000 | All | ✗ | ✗ | ✗ | 59.0 | 4.00 |
| Post-SH (Guo et al., 2017) | 1,000 | Post Processing | ✓ | ✗ | ✗ | 118.8 | 0.60 |
| Post-SH (Guo et al., 2017) | 500 | Post Processing | ✓ | ✗ | ✗ | 166.8 | 0.30 |
| Post-SH+k-SVD | 1,000 | Post Processing | ✓ | ✗ | ✗ | 78.0 | 0.60 |
| Post-SH+k-SVD | 500 | Post Processing | ✓ | ✗ | ✗ | 103.5 | 0.30 |
| Anchor & Transform | 1,000 | Dynamic | ✓ | ✓ | ✗ | 72.0 | 0.44 |
| | 500 | Dynamic | ✓ | ✓ | ✗ | 74.0 | 0.26 |
| | 1,000 | Frequency | ✓ | ✓ | ✗ | 77.0 | 0.45 |
| | 100 | Frequency | ✓ | ✓ | ✓ | **70.0** | **0.05** |

500/1000 reaches a good perplexity/compression trade-off: we reach within 2 points perplexity with 5× reduction in parameters and within 7 points perplexity with 10× reduction. Using AWD-LSTM, ANT with 1,000 dynamic basis vectors is able to compress parameters by 10× while achieving 72.0 perplexity. Incorporating domain knowledge allows us to further compress the parameters by *another* 10× and achieve 70.0 perplexity, which results in 100× total compression.

On WikiText-103, we train all approaches using sampled softmax (Bengio & Senecal, 2008) (as the vocabulary is relatively large) for 500,000 steps. To best of our knowledge, we could not find existing literature on compressing language models on WikiText-103[2]. We tried general compression techniques like low rank tensor and tensor train factorization (Grachev et al., 2019), but these did not scale. As an alternative, we consider a **Hash Embed** baseline that retains the frequent $k$ words and hashes the remaining words into 1,000 OOV buckets (Svenstrup et al., 2017). We vary $k \in \{1 \times 10^5, 5 \times 10^4, 1 \times 10^4\}$. The results in Table 3 show that we can reach within 3 points perplexity with ~ 16× reduction in parameters and within 10 points perplexity with 100× reduction, outperforming the frequency and hashing baselines. We observe that the performance improvement of ANT over post-processing compression methods (Post-SH and Post-SH+k-SVD) is larger on WikiText-103 as compared to PTB, demonstrating that our end-to-end sparse embedding method is particularly suitable for tasks with large vocabularies.

**Sparse transformations learned:** We can visualize the important transformations (large entries) learned between anchors and non-anchors. Using anchors initialized by frequency and using domain knowledge from WordNet and co-occurrence, we show the learnt associations in Table 4 after training AWD-LSTM on PTB. On the left we show the most associated non-anchor words for a given anchor word such as *year* or *stock* and we find that the induced non-anchor words are highly plausible: *stock* accurately contributes to the representations for *bonds, certificates, securities*, and so on. On the right, we show the largest (non-anchor, anchor) pairs learned. Again, we find related concepts such as *(when, how)*, *(billion, trillion)*, and *(government, administration)* (more examples in Table 4).

---

[2]While Baevski & Auli (2018) adapt embedding dimensions according to word frequencies, their goal is not compression and use 44.9M (dense) parameters in their adaptive embedding layer, while we use as less as 0.4M (100x less). Also they obtained results using a Transformer with 250M params while AWD-LSTM uses 130M.

Table 3: Language modeling on WikiText-103. We reach within 3 perplexity with ~ 16× reduction and within 10 perplexity with 100× reduction, outperforming frequency and hashing baselines.

| Method | $|A|$ | Init $A$ | Sparse $\mathbf{T}$ | RELU($\mathbf{T}$) | Domain | Ppl | # Emb (M) |
|---|---|---|---|---|---|---|---|
| AWD-LSTM (Merity et al., 2017) | 267, 735 | All | ✗ | ✗ | ✗ | 35.2 | 106.8 |
| Hash Embed (Svenstrup et al., 2017) | 100, 000 | Frequency | ✗ | ✗ | ✗ | 40.6 | 40.4 |
| Hash Embed (Svenstrup et al., 2017) | 50, 000 | Frequency | ✗ | ✗ | ✗ | 52.5 | 20.4 |
| Hash Embed (Svenstrup et al., 2017) | 10, 000 | Frequency | ✗ | ✗ | ✗ | 70.2 | 4.4 |
| Post-SH (Guo et al., 2017) | 1, 000 | Post Processing | ✓ | ✗ | ✗ | 764.7 | 5.7 |
| Post-SH (Guo et al., 2017) | 500 | Post Processing | ✓ | ✗ | ✗ | 926.8 | 2.9 |
| Post-SH+k-SVD | 1, 000 | Post Processing | ✓ | ✗ | ✗ | 73.7 | 5.7 |
| Post-SH+k-SVD | 500 | Post Processing | ✓ | ✗ | ✗ | 148.3 | 2.9 |
| | 1, 000 | Dynamic ($\lambda = 10^{-6}$) | ✓ | ✓ | ✗ | **38.4** | 6.5 |
| Anchor & Transform | 1, 000 | Dynamic ($\lambda = 10^{-5}$) | ✓ | ✓ | ✗ | 39.7 | 3.1 |
| | 500 | Dynamic ($\lambda = 10^{-6}$) | ✓ | ✓ | ✗ | 48.8 | 1.4 |
| | 500 | Dynamic ($\lambda = 10^{-5}$) | ✓ | ✓ | ✗ | 54.2 | **0.4** |

Table 4: Word association results after training language models with ANT on the word-level PTB dataset. Left: the non-anchor words most induced by a given anchor word. Right: the largest (non-anchor, anchor) entries learnt in $\mathbf{T}$ after sparse regularization.

| Largest word pairs |
|---|
| *trading, brokerage* |
| *stock, junk* |
| *year, summer* |
| *york, angeles* |
| *year, month* |
| *government, administration* |
| *two, nine* |

| Anchor words | Non-anchor words |
|---|---|
| *year* | *august, night, week, month, monday, summer, spring* |
| *stock* | *bonds, certificates, debt, notes, securities, mortgages* |

## 4.3 DISCUSSION

Here we list some general observations regarding the importance of various design decision in ANT:

1) **Sparsity is important:** Baseline methods that only perform low-rank compression with dense factors (e.g. LR) tend to suffer in performance while using many parameters, while ANT with sparsity regularization retains performance with much better compression.

2) **Proximal GD is essential:** To achieve desired sparsity when training with mini-batches, the use of proximal GD was crucial. With default subgradient descent we did not observe sparsification of $\mathbf{T}$.

3) **Choice of** $A$**:** Performance is robust wrt choice of $A$. We give a detailed discussion in Appendix A and provide results on clustering initializations. In general, while frequency and clustering work better, using a dynamic basis still performs well, especially when combined with domain knowledge. This implies that when the user has more information about the discrete objects (e.g. having a good representation space like GloVe to perform clustering), then the user should do so. However, for a new set of discrete objects, using random basis embeddings with sparsity also works well.

4) **RELU:** Passing $\mathbf{T}$ through RELU is important to reduce redundancy in each row entry.

5) **Mixture:** Using a mixture of anchors and transformations also achieves stronger performance than existing baselines, although the larger number of transformations increases the parameters.

6) **Incorporating domain knowledge (DK):** DK from WordNet and co-occurrence statistics help to further reduce the total (non-zero) embedding parameters while maintaining performance.

## 5 CONCLUSION

This paper presented Anchor & Transform (ANT) to learn sparse representations of discrete objects by 1) learning a small set of *anchor embeddings* and 2) learning a *sparse transformation* from anchors to all objects. ANT is scalable, flexible, end-to-end trainable, and allows the user to easily incorporate domain knowledge about object relationships. On experiments spanning text classification and language modeling, ANT demonstrates strong performance with respect to accuracy and sparsity, outperforming existing compression approaches.

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

APPENDIX

## A  LEARNING THE ANCHOR EMBEDDINGS A

Here we provide several other strategies for initializing the anchor embeddings:

- Sparse lasso and variational dropout (Chen et al., 2019). Given the strong performance of sparse lasso and variational dropout as vocabulary selection methods (Chen et al., 2019), it would be interesting to use sparse lasso/variational dropout to first select the important task-specific words before jointly learning their representations and their transformations to other words. However, sparse lasso and variational dropout require first training a model to completion unlike frequency and clustering based vocabulary selection methods that can be performed during data preprocessing.
- Coresets involve constructing a reduced data set which can be used as proxy for the full data set, with provable guarantees such that the same algorithm run on the coreset and the full data set gives approximately similar results (Phillips, 2016; Har-Peled & Mazumdar, 2018). Coresets can be approximately computed quickly (Bachem et al., 2017) and can be used to initialize the set of anchors $A$.

In general, there is a trade-off between how quickly we can choose the anchor objects and their performance. Randomly picking anchor objects (which is equivalent to initializing the anchor embeddings with dynamic basis vectors) becomes similar to learning a low-rank factorization of the embedding matrix (Nam & Quoc, 2017; Sedov & Yang, 2018), which works well for general cases but can be improved for task-specific applications or with domain knowledge. Stronger vocabulary selection methods like variational dropout and group lasso would perform better but takes significantly longer time to learn. We found that intermediate methods such as frequency, clustering, with WordNet/co-occurrence information works well while ensuring that the preprocessing and training stages are relatively quick.

In Appendix F we provide more results for different initialization strategies including those based on clustering initializations. In general, performance is robust with respect to the choice of $A$ among the ones considered (i.e. random, frequency, and clustering). While frequency and clustering work better, using a set of dynamic basis embeddings still gives strong performance, especially when combined with domain knowledge from WordNet and co-occurrence statistics. This implies that when the user has more information about the discrete objects (e.g. having a good representation space to perform clustering), then the user should do so. However, for a completely new set of discrete objects, simply using low-rank basis embeddings with sparsity also work well.

## B  CONNECTION TO SPARSE DICTIONARY LEARNING AND SPARSE RECOVERY

Consider the modified subproblem where instead of jointly optimizing over both $\mathbf{A}$ and $\mathbf{T}$, we assume that the anchor embeddings $\mathbf{A}^*$ are given. In practice, this could imply that we first select an important subset of anchor words and then embed them with a pretrained representation space such as GloVe embeddings (Pennington et al., 2014). Then, the problem is to learn a sparse transformation $\mathbf{T}$ from these pretrained anchor embeddings to the remaining embeddings (conceptually similar to recent work on reconstructing pretrained word embeddings using low rank tensors (Nam & Quoc, 2017; Sedov & Yang, 2018) or codebook learning (Shu & Nakayama, 2018; Chen et al., 2018)). Formally, the problem becomes

$$\min_{\mathbf{T}} \|\mathbf{T}\|_0 \quad \text{subject to } \mathbf{E}^* = \mathbf{T}\mathbf{A}^*, \tag{6}$$

which is connected to the sparse dictionary learning problem (Awasthi & Vijayaraghavan, 2018) where we aim to learn an unknown dictionary $\mathbf{A}^*$ and a sparse representation $\mathbf{T}$ that generated data $\mathbf{E} = \mathbf{T}\mathbf{A}^*$. As with existing results on recovering dictionaries in the over-complete setting, we assume that matrix $\mathbf{A}^*$ satisfies the *Restricted Isometry Property (RIP)* condition:

**Definition 1.** *A matrix $\Phi$ satisfies the Restricted Isometry Property (RIP) with isometry constant $\delta_k$ if for all $k$-sparse vectors $\mathbf{x}$, we have*

$$(1 - \delta_k)\|\mathbf{x}\|_2^2 \leq \|\Phi\mathbf{x}\|_2^2 \leq (1 + \delta_k)\|\mathbf{x}\|_2^2.$$

Suppose (i) $\mathbf{A}^{*T}$ obeys the *Restricted Isometry Property (RIP)*, (ii) the true transformation matrix $\mathbf{T}^*$ is $k$-*row sparse* (each row of $\mathbf{T}^*$ has at most $k$ non-zero entries), and (iii) the true embedding $\mathbf{E}^*$

was known, then recovering $\mathbf{T}^*$ from $\mathbf{A}$ and $\mathbf{E}^*$ is possible by using the $\ell_1$ regularization (similar to equation 2).

Then, a classical result by Candes & Tao (2005); Candès (2008) states if $\mathbf{y} = \Phi\mathbf{x}^*$ for some $k$-sparse vector $\mathbf{x}^*$, then the solution to

$$\min_{\mathbf{x}} \ \|\mathbf{x}\|_0 \quad \text{subject to } \mathbf{y} = \Phi\mathbf{x} \tag{7}$$

is equivalent to the solution given by the $\ell_1$ problem

$$\min_{\mathbf{x}} \ \|\mathbf{x}\|_1 \quad \text{subject to } \mathbf{y} = \Phi\mathbf{x} \tag{8}$$

when $\delta_{2k} \leq \sqrt{2} - 1$.

In our setting, we aim to solve the constrained optimization problem

$$\min_{\mathbf{T}} \ \|\mathbf{T}\|_0 \quad \text{subject to } \mathbf{E}^* = \mathbf{T}\mathbf{A}^*, \tag{9}$$

where each row in $\mathbf{T}$ may be minimized independently of the rest. Assuming that $\mathbf{E}^* = \mathbf{T}^*\mathbf{A}$ for some $k$-row sparse matrix $\mathbf{T}^*$, applying the result by Candes & Tao (2005); Candès (2008) to each row and exploiting the assumption that $\mathbf{A}^{*T}$ is RIP gives

$$\min_{\mathbf{T}} \ \|\mathbf{T}\|_1 \quad \text{subject to } \mathbf{E}^* = \mathbf{T}\mathbf{A}^*. \tag{10}$$

which is equivalent to $\ell_1$ regularization in equation 2 when written in Lagrange form. Therefore, $\ell_1$-regularization (like equation 2) provably results in sparse entries in $\mathbf{T}$.

## C   EFFICIENT LEARNING AND INFERENCE

The naive method for learning $\mathbf{E}$ from anchor embeddings $\mathbf{A}$ and the sparse transformations $\mathbf{T}$ still scales linearly with $|V| \times d$. Here we describe some tips on how to perform efficient learning and inference of the anchor embeddings $\mathbf{A}$ and the sparse transformations $\mathbf{T}$:

- Store $\mathbf{T}$ as a sparse matrix by only storing its non-zero entries and indices. From our experiments, we have shown that $\text{nnz}(\mathbf{T}) << |V| \times d$ which makes storage efficient.
- For inference, use sparse matrix multiply as supported in TensorFlow and PyTorch to compute $\mathbf{E} = \mathbf{T}\mathbf{A}$ (or its non-linear extensions). This decreases the running time from scaling by $|V| \times d$ to only scaling as a function of $\text{nnz}(\mathbf{T})$. For training, using inbuilt sparse representation of most deep learning frameworks like PyTorch or Tensorflow is not optimal, as they do not support changing non-zero locations in sparse matrix and apriori its not easy to find optimal set of non-zero locations.
- During training, instead, implicitly construct $\mathbf{E}$ from its anchors and transformations. In fact, we can do better: instead of constructing the entire $\mathbf{E}$ matrix to embed a single datapoint $\mathbf{x} \in \mathbb{R}^{1 \times |V|}$, we can instead first *index* $\mathbf{x}$ into $\mathbf{T}$, i.e. $\mathbf{x}\mathbf{T} \in \mathbb{R}^{1 \times |A|}$ before performing a sparse matrix multiplication with $\mathbf{A}$, i.e. $(\mathbf{x}\mathbf{T})\mathbf{A} \in \mathbb{R}^{1 \times d}$. We are essentially taking advantage of the *associative* property of matrix multiplication and the fact that $\mathbf{x}\mathbf{T}$ is a simple indexing step and $(\mathbf{x}\mathbf{T})\mathbf{A}$ is an effective sparse matrix multiplication. To enable fast row slicing into sparse matrix, we just storing the matrix in adjacency list or CSOO format. (We move away from CSR as adding/deleting a non-zero location is very expensive.) When gradient comes back, only update the corresponding row in $\mathbf{T}$. The gradient will be sparse as well due to the L1-prox operator.
- Above trick solves the problem for tasks where embedding is used only at the input, e.g. classification. For tasks like language model, where embedding is used at output as well one can also use above mentioned trick with speedup techniques like various softmax sampling techniques (Bengio & Senecal, 2008; Mikolov et al., 2013) or noise-contrastive estimation (Gutmann & Hyvrinen, 2010; Mnih & Teh, 2012), which will be anyway used for large vocabulary sizes. To elaborate, consider the case of sampled softmax (Bengio & Senecal, 2008). We normally generate the negative sample indices, and then we can first *index* into $\mathbf{T}$ using the true and negative indices before performing sparse matrix multiplication with $\mathbf{A}$. This way we do not have to instantiate entire $\mathbf{E}$ by expensive matrix multiplication.
- When training is completed, only store the non-zero entries of $\mathbf{T}$ or store $\mathbf{T}$ as a sparse matrix to reconstruct $\mathbf{E}$ for inference.

- To save time when initializing the anchor embeddings and incorporating domain knowledge, precompute the necessary statistics such as frequency statistics, co-occurrence statistics, and object relation statistics. We use a small context size of 10 to measure co-occurrence of two words to save time. When using WordNet to discover word relations, we only search for immediate relations between words instead of propagating relations across multiple steps (although this could further improve performance).
- In order to incorporate domain knowledge in the sparsity structure, we again store $1 - \mathbf{S}(G)$ using sparse matrices. Recall that $\mathbf{S}(G)$ has an entry equal to 1 for entries representing unrelated objects that should be $\ell_1$-penalized, which makes $\mathbf{S}(G)$ quite dense since most anchor and non-anchor objects are unrelated. Hence we store $1 - \mathbf{S}(G)$ instead which consists few non-zero entries only at (non-anchor, anchor) entries for related objects. Element-wise multiplications are also replaced by sparse element-wise multiplications when computing $\mathbf{T} \odot \mathbf{S}(G)$ and $\mathbf{T} \odot (1 - \mathbf{S}(G))$.
- Finally, even if we want to utilize our ANTframework with full softmax in language model, it is possible without blowing up memory requirements. In particular, let $\mathbf{g} \in \mathbf{R}^{1 \times |V|}$ be the incoming gradient from cross-entropy loss and $\mathbf{h} \in \mathbf{R}^{d \times 1}$ be the vector coming from layers below, like LSTM. The gradient update is then

$$\mathbf{T} \leftarrow \text{PROX}_{\eta\lambda}(\mathbf{T} - \eta\mathbf{g}(\mathbf{Ah})^T) \tag{11}$$

The main issue is computing the huge $|V| \times |A|$ outer product as an intermediate step which will be dense. However, note that incoming gradient $\mathbf{g}$ is basically a softmax minus an offset corresponding to correct label. This should only have large values for a small set of words and small for others. If we carefully apply the L1-prox operator earlier, which is nothing but a soft-thresholding, we can make this incoming gradient sparse very sparse. Thus we need to only calculate a much smaller sized outer product and touch a small number of rows in $\mathbb{T}$. Thus, making the approach feasible.

## D  GENERALITY OF ANT

We show that under certain structural assumptions on the anchor embeddings and transformation matrices, ANT reduces to the following task-specific methods for learning sparse representations. This implies that ANT is indeed a general framework that unifies some of the work on sparse representation learning done independently in different research areas.

**Frequency-based vocabulary selection** (Luong et al., 2015; Chen et al., 2016b): Initialize $A$ with the $|A|$ most frequent objects and set $\mathbf{T}_{a,a} = 1$ for all $a \in A$, $\mathbf{T} = 0$ otherwise. Then $\mathbf{E} = \mathbf{TA}$ consists of embeddings of the $|A|$ most frequent objects with zero embeddings for all others. During training, gradients are used to update $\mathbf{A}$ but not $\mathbf{T}$ (i.e. only embeddings for frequent objects are learned). By changing the selection of $A$, ANT also reduces to other vocabulary selection methods such as TF-IDF (Ramos, 1999), Group Lasso (Wen et al., 2016), and variational dropout (Chen et al., 2019)

**Low-rank factorization** (Acharya et al., 2018; Markovsky, 2011; Grachev et al., 2019): Initialize $A$ by a mixture of random basis embeddings (just 1 anchor per set) $\mathbf{A}_1, ..., \mathbf{A}_M \in \mathbb{R}^{1 \times d}$ and do not enforce any sparsity on the transformations $\mathbf{T}_1, ..., \mathbf{T}_M \in \mathbb{R}^{|V| \times 1}$. If we further restrict ourselves to only linear combinations $\mathbf{E} = \sum_{m=1}^{M} \mathbf{T}_m \mathbf{A}_m$, this is equivalent to implicitly learning the $M$ low rank factors $\mathbf{a}_1, ..., \mathbf{a}_M, \mathbf{t}_1, ..., \mathbf{t}_M$ that reconstruct embedding matrices of rank at most $M$.

**Compositional code learning** (Shu & Nakayama, 2018; Chen et al., 2018): Initialize $A$ by a mixture of random basis embeddings $\mathbf{A}_1, ..., \mathbf{A}_M$, initialize transformations $\mathbf{T}_1, ..., \mathbf{T}_M$, and apply a linear combination $\mathbf{E} = \sum_{m=1}^{M} \mathbf{T}_m \mathbf{A}_m$. For sparsity regularization, set row $i$ of $\mathbf{S}(G)_{mi}$ as a reverse one-hot vector with entry $d_{mi} = 0$ and all else 1. In other words, index $d_{mi}$ of row row $\mathbf{T}_{mi}$ is not regularized, and all other entries are $\ell_1$-regularized with extremely high $\lambda$ such that row $\mathbf{T}_{mi}$ essentially becomes an one-hot vector with dimension $d_{mi} = 1$. This results in learning a *codebook* where each object in $V$ is mapped to *only one* anchor in each mixture.

Therefore, ANT encompasses several popular methods for learning sparse representations, and gives further additional flexibility in defining various initialization strategies, applying nonlinear mixtures of transformations, and incorporating domain knowledge via object relationships.

Table 5: Table of hyperparameters for text classification experiments on AG-News, DBPedia, Sogou-News, and Yelp-review datasets. All text classification experiments use the same base CNN model with the exception of different output dimensions (classes in the dataset): 4 for AG-News, 14 for DBPedia, 5 for Sogou-News, and 5 for Yelp-review.

| Model | Parameter | Value |
|---|---|---|
| | Embedding dim | 256 |
| | Filter sizes | $[3, 4, 5]$ |
| | Num filters | 100 |
| | Filter strides | $[1, 1]$ |
| | Filter padding | valid |
| | Pooling strides | $[1, 1]$ |
| | Pooling padding | valid |
| | Loss | cross entropy |
| model | Dropout | 0.5 |
| | Batch size | 256 |
| | Max seq length | 100 |
| | Num epochs | 200 |
| | Activation | ReLU |
| | Optimizer | Adam |
| | Learning rate | $5 \times 10^{-3}$ |
| | Learning rate decay | $1 \times 10^{-5}$ |
| | Start decay | 40 |

# E    EXPERIMENTAL DETAILS

Here we provide more details for our experiments including hyperparameters used, design decisions, and comparison with baseline methods.

## E.1    TEXT CLASSIFICATION

**Base CNN model:** For all text classification experiments, the base model is a CNN (Lecun et al., 1998) with layers of 2D convolutions and 2D max pooling, before a dense layer to the output softmax. The code was adapted from `https://github.com/wenhuchen/Variational-Vocabulary-Selection` and the architecture hyperparameters are provided in Table 5. The only differences are the output dimensions which is 4 for AG-News, 14 for DBPedia, 5 for Sogou-News, and 5 for Yelp-review.

**Anchor:** We experiment with dynamic, frequency, and clustering initialization strategies. The number of anchors $|A|$ is a hyperparameter that is selected using the validation set. The range of $|A|$ is in $\{10, 20, 50, 80, 100, 500, 1,000\}$. Smaller values of $|A|$ allows us to control for fewer anchors and smaller transformation matrix $\mathbf{T}$ at the expense of performance.

**Transformation:** We experiment with sparse linear transformations for $\mathbf{T}$. $\lambda$ is a hyperparameter that is selected using the validation set. Larger values of $\lambda$ allows us to control for more sparse entries in $\mathbf{T}$ at the expense of performance. For experiments on dynamic mixtures, we use a softmax-based nonlinear combination $\mathbf{E} = \sum_{m=1}^{M} \mathrm{softmax}(\mathbf{T}_m)\mathbf{A}_m$ where softmax is performed over the rows of $\mathbf{T}_m$. Note that applying a softmax activation to the rows of $\mathbf{T}_m$ makes all entries dense so during training, we store $\mathbf{T}_m$ as sparse matrices (which is efficient since $\mathbf{T}_m$ has few non-zero entries) and *implicitly* reconstruct $\mathbf{E}$.

**Domain knowledge:** When incorporating domain knowledge in ANT, we use both WordNet and co-occurrence statistics. For WordNet, we use the public WordNet interface provided by NLTK `http://www.nltk.org/howto/wordnet.html`. For each word we search for its immediate related words among its hypernyms, hyponyms, synonyms, and antonyms. This defines the relationship graph. For co-occurrence statistics, we define a co-occurrence context size of 10 on the training data. Two words are defined to be related if they co-occur within this context size.

**A note on baselines:** Note that the reported results on SparseVD and SparseVD-Voc (Chirkova et al., 2018) have a different embedding size: 300 instead of 256. This is because they use pre-trained word2vec or GloVe embeddings to initialize their model before compression is performed.

Table 6: Table of hyperparameters for language modeling experiments using LSTM on PTB dataset.

| Model | Parameter | Value |
|---|---|---|
| | Embedding dim | 200 |
| | Num hidden layers | 2 |
| | Hidden layer size | 200 |
| | Output dim | $10,000$ |
| | Loss | cross entropy |
| | Dropout | 0.4 |
| | Word embedding dropout | 0.1 |
| | Input embedding dropout | 0.4 |
| | LSTM layers dropout | 0.25 |
| | Weight dropout | 0.5 |
| model | Weight decay | $1.2 \times 10^{-6}$ |
| | Activation regularization | 2.0 |
| | Temporal activation regularization | 1.0 |
| | Batchsize | 20 |
| | Max seq length | 70 |
| | Num epochs | 500 |
| | Activation | ReLU |
| | Optimizer | SGD |
| | Learning rate | 30 |
| | Gradient clip | 0.25 |
| | Learning rate decay | $1 \times 10^{-5}$ |
| | Start decay | 40 |

### E.2 LANGUAGE MODELING ON PTB

**Base LSTM model:** Our base model is a 2 layer LSTM with an embedding size of 200 and hidden layer size of 200. The code was adapted from `https://github.com/salesforce/awd-lstm-lm` and the full table of hyperparameters is provided in Table 6.

**Base AWD-LSTM model:** In addition to experiments on an vanilla LSTM model as presented in the main text, we also performed experiments using a 3 layer AWD-LSTM with an embedding size of $400$ and hidden layer size of $1,150$. The full hyperparameters used can be found in Table 7.

**Anchor:** We experiment with dynamic, frequency, and clustering initialization strategies. The number of anchors $|A|$ is a hyperparameter that is selected using the validation set. The range of $|A|$ is in $\{10, 20, 50, 80, 100, 500, 1,000\}$. Smaller values of $|A|$ allows us to control for fewer anchors and smaller transformation matrix $\mathbf{T}$ at the expense of performance.

**Domain knowledge:** When incorporating domain knowledge in ANT, we use both WordNet and co-occurrence statistics. For WordNet, we use the public WordNet interface provided by NLTK `http://www.nltk.org/howto/wordnet.html`. For each word we search for its immediate related words among its hypernyms, hyponyms, synonyms, and antonyms. This defines the relationship graph. For co-occurrence statistics, we define a co-occurrence context size of 10 on the training data. Two words are defined to be related if they co-occur within this context size.

**A note on baselines:** We also used some of the baseline results as presented in Grachev et al. (2019). Their presented results differ from our computations in two aspects: they include the LSTM parameters on top of the embedding parameters, and they also count the embedding parameters twice since they do not perform weight tying (Press & Wolf, 2016) (see equation (6) of Grachev et al. (2019)). To account for this, the results of SparseVD and SparseVD-Voc (Chirkova et al., 2018), as well as the results of various LR- and TR- low rank compression methods (Grachev et al., 2019) were modified by subtracting off the LSTM parameters ($200 \times 200 \times 16$). This is derived since each of the 8 weight matrices $W_{i,f,o,c}, U_{i,f,o,c}$ in an LSTM layer is of size $200 \times 200$, and there are a 2 LSTM layers. We then divide by two to account for weight tying. In the main text, we compared with the *strongest* baselines as reported in Grachev et al. (2019): these were the methods that performed low rank decomposition on both the input embedding ($|V| \times d$), output embedding ($d \times |V|$), and intermediate hidden layers of the model. For full results, please refer to Grachev et al. (2019).

Note that the reported results on SparseVD and SparseVD-Voc (Chirkova et al., 2018) have a different embedding size and hidden layer size of 256 instead of 200, although these numbers are close enough

Table 7: Table of hyperparameters for language modeling experiments using AWD-LSTM on PTB dataset.

| Model | Parameter | Value |
|---|---|---|
| | Embedding dim | 400 |
| | Num hidden layers | 3 |
| | Hidden layer size | $1,150$ |
| | Output dim | $10,000$ |
| | Loss | cross entropy |
| | Dropout | 0.4 |
| | Word embedding dropout | 0.1 |
| | Input embedding dropout | 0.4 |
| | LSTM layers dropout | 0.25 |
| | Weight dropout | 0.5 |
| model | Weight decay | $1.2 \times 10^{-6}$ |
| | Activation regularization | 2.0 |
| | Temporal activation regularization | 1.0 |
| | Batchsize | 20 |
| | Max seq length | 70 |
| | Num epochs | 500 |
| | Activation | ReLU |
| | Optimizer | SGD |
| | Learning rate | 30 |
| | Gradient clip | 0.25 |
| | Learning rate decay | $1 \times 10^{-5}$ |
| | Start decay | 40 |

for fair comparison. In our experiments we additionally implemented an LSTM with an embedding size of 256 and hidden layer size of 256 so that we can directly compare with their reported numbers.

For baselines that perform post-processing compression of the embedding matrix, Post-SH (post-processing using sparse hashing) (Guo et al., 2017) and Post-SH+k-SVD (improving sparse hashing using k-SVD) (Guo et al., 2017; Awasthi & Vijayaraghavan, 2018), we choose two settings: the first using 500 anchors and 10 nearest neighbors to these anchor points, and the second using $1,000$ anchors and 20 nearest neighbors. The first model uses $500 \times d + |V| \times 10$ non-zero embedding parameters while the second model uses $1,000 \times d + |V| \times 20$ parameters. For AWD-LSTM on PTB, this is equivalent to 0.3M and 0.6M embedding parameters respectively which is comparable to the number of non-zero parameters used by our method.

### E.3 LANGUAGE MODELING ON WIKITEXT-103

**Base AWD-LSTM model:** Our base model is a 4 layer AWD-LSTM with an embedding size of 400 and hidden layer size of $2,500$. The code was adapted from `https://github.com/salesforce/awd-lstm-lm` and the hyperparameters used can be found in Table 8.

**A note on baselines:** While Baevski & Auli (2018) adapt embedding dimensions according to word frequencies, their goal is not to compress embedding parameters and they use 44.9M (dense) parameters in their adaptive embedding layer, while we use only 2M. Their embedding parameters are calculated by their reported bucket sizes and embedding sizes (three bands of size 20K ($d = 1024$), 40K ($d = 256$) and 200K ($d = 64$)). Their perplexity results are also obtained using a Transformer model with 250M params while our AWD-LSTM model uses 130M params.

For the **Hash Embed** baseline that retains the frequent $k$ words and hashes the remaining words into $1,000$ OOV buckets (Svenstrup et al., 2017), We vary $k \in \{1 \times 10^5, 5 \times 10^4, 1 \times 10^4\}$.

## F   MORE RESULTS

In the following sections we provide additional results on learning sparse representations of discrete objects using ANT.

### F.1 TEXT CLASSIFICATION

**Results:** We report additional text classification results on DBPedia, Sogou-News, and Yelp-review in Table 9. Our approach with different initializations and domain knowledge achieves within 1%

Table 8: Table of hyperparameters for language modeling experiments using AWD-LSTM on WikiText-103 dataset.

| Model | Parameter | Value |
|---|---|---|
| | Embedding dim | 400 |
| | Num hidden layers | 4 |
| | Hidden layer size | $2,500$ |
| | Output dim | $267,735$ |
| | Loss | cross entropy |
| | Dropout | 0.1 |
| | Word embedding dropout | 0.0 |
| | Input embedding dropout | 0.1 |
| | LSTM layers dropout | 0.1 |
| | Weight dropout | 0.0 |
| model | Weight decay | 0.0 |
| | Activation regularization | 0.0 |
| | Temporal activation regularization | 0.0 |
| | Batchsize | 32 |
| | Max seq length | 140 |
| | Num epochs | 14 |
| | Activation | ReLU |
| | Optimizer | SGD |
| | Learning rate | 30 |
| | Gradient clip | 0.25 |
| | Learning rate decay | $1 \times 10^{-5}$ |
| | Start decay | 40 |

accuracy with $21\times$ fewer parameters on DBPedia, within $1\%$ accuracy with $10\times$ fewer parameters on Sogou-News, and within $2\%$ accuracy with $22\times$ fewer parameters on Yelp-review.

**Different initialization strategies:** Here we also presented results across different initialization strategies and find that while those based on frequency and clustering work better, using a set of dynamic basis embeddings still gives strong performance, especially when combined with domain knowledge from WordNet and co-occurrence statistics. This implies that when the user has more information about the discrete objects (e.g. having a good representation space to perform clustering), then the user should do so. However, for a completely new set of discrete objects, simply using low-rank basis embeddings with sparsity also work well.

## F.2 LANGUAGE MODELING

**Results:** We report additional language modeling results using AWD-LSTM on PTB in Table 10. ANT with $1,000$ dynamic basis vectors is able to compress the embedding parameters by $10\times$ while achieving 72.0 test perplexity. By incorporating domain knowledge, we further compress the embedding parameters by *another* $10\times$ and achieve 70.0 test perplexity, which results in $100\times$ total compression as compared to the baseline.

Table 9: More text classification results on DBPedia (top), Sogou-News (middle), and Yelp-review (bottom). Domain knowledge is derived from WordNet and co-occurrence statistics. Our approach with different initializations and domain knowledge achieves within $1\%$ accuracy with $21\times$ fewer parameters on DBPedia, within $1\%$ accuracy with $10\times$ fewer parameters on Sogou-News, and within $2\%$ accuracy with $22\times$ fewer parameters on Yelp-review. Acc: accuracy, # Emb: # (non-zero) embedding parameters.

| Method | $|A|$ | Init $A$ | Sparse $\mathbf{T}$ | RELU($\mathbf{T}$) | Domain | Acc (%) | # Emb (M) |
|---|---|---|---|---|---|---|---|
| CNN (Zhang et al., 2015) | $563,355$ | All | ✗ | ✗ | ✗ | 98.3 | 144.0 |
| Sparse Code (Chen et al., 2016b) | 100 | Frequency | ✓ | ✗ | ✗ | 96.7 | 39.0 |
| | 80 | Cluster | ✓ | ✓ | ✗ | 98.1 | 30.0 |
| Anchor & Transform | 100 | Dynamic | ✓ | ✓ | ✗ | 98.2 | 28.0 |
| | 50 | Frequency | ✓ | ✓ | ✓ | 97.3 | 18.0 |
| | 20 | Frequency | ✓ | ✓ | ✓ | **97.2** | **7.0** |

| Method | $|A|$ | Init $A$ | Sparse $\mathbf{T}$ | RELU($\mathbf{T}$) | Domain | Acc (%) | # Emb (M) |
|---|---|---|---|---|---|---|---|
| CNN (Zhang et al., 2015) | $254,495$ | All | ✗ | ✗ | ✗ | 94.0 | 65.0 |
| Sparse Code (Chen et al., 2016b) | 100 | Frequency | ✓ | ✗ | ✗ | 92.0 | 6.0 |
| | 50 | Cluster | ✓ | ✓ | ✗ | **93.0** | **5.0** |
| Anchor & Transform | 80 | Cluster | ✓ | ✓ | ✗ | 93.1 | 9.0 |
| | 100 | Dynamic | ✓ | ✓ | ✗ | 93.2 | 6.0 |
| | 50 | Frequency | ✓ | ✓ | ✓ | 92.0 | **5.0** |

| Method | $|A|$ | Init $A$ | Sparse $\mathbf{T}$ | RELU($\mathbf{T}$) | Domain | Acc (%) | # Emb (M) |
|---|---|---|---|---|---|---|---|
| CNN (Zhang et al., 2015) | $252,712$ | All | ✗ | ✗ | ✗ | 56.2 | 65.0 |
| Sparse Code (Chen et al., 2016b) | 100 | Frequency | ✓ | ✗ | ✗ | 54.0 | 14.0 |
| | 80 | Cluster | ✓ | ✓ | ✗ | 56.2 | 8.0 |
| Anchor & Transform | 50 | Dynamic | ✓ | ✓ | ✗ | 55.7 | 6.0 |
| | 50 | Dynamic | ✓ | ✓ | ✗ | 56.0 | 6.0 |
| | 50 | Frequency | ✓ | ✓ | ✓ | **54.7** | **3.0** |

Table 10: More language modeling results using AWD-LSTM on Penn Treebank. Using domain knowledge to infer sparse structures helps to reduce the embedding parameters by $100\times$. Ppl: per-plexity, # Emb: # (non-zero) embedding parameters.

| Method | $|A|$ | Init $A$ | Sparse $\mathbf{T}$ | RELU($\mathbf{T}$) | Domain | Ppl | # Emb (M) |
|---|---|---|---|---|---|---|---|
| AWD-LSTM (Merity et al., 2017) | $10,000$ | All | ✗ | ✗ | ✗ | 59.0 | 4.00 |
| | $1,000$ | Dynamic | ✓ | ✓ | ✗ | 72.0 | 0.44 |
| Anchor & Transform | $1,000$ | Frequency | ✓ | ✓ | ✗ | 77.0 | 0.45 |
| | 100 | Frequency | ✓ | ✓ | ✓ | **70.0** | **0.05** |

