# OpenReview forum: "Anchor & Transform: Learning Sparse Representations of Discrete Objects"
_ICLR.cc/2020/Conference — Reject_

### Official Review · AnonReviewer3 · 2019-10-23
**Official Blind Review #3**

**Rating:** 6

**Review:**

The paper describes a "layer" that aims at producing embeddings for discrete objects by using fewer parameters than classical embeddings layers. Indeed, the model proposes, instead of learning an embedding matrix of size VxN, to learn a matrix of embeddings of anchors (AxN) and a transformation matrix (VxA) such that the embedding of any object can be found by multiplying A with T. On top of that, they propose different regularization techniques to improve the quality of the learned embeddings, and particularly a proximal gradient method over a L1 normalization on T to reduce the number of parameters. They propose also different ways to initialize A and also a method for incorporating a priori information (e.g knowledge) into the model.  They evaluate this model on different tasks: text classification and language modeling and show that they can achieve good performance while using fewer parameters than Sota methods.

First of all, the paper is well written, and the description is very detailed and understandable. It was a pleasure to read such a paper!

One point which is unclear is the interest of using such a method, and more precisely in which cases, this method can be useful. Indeed, the overall number of parameters of ANT is AxN + VxA (N being the size of the embeddings, A the number of anchors and V the size of the vocabulary) while classical methods are VxN parameters. Said otherwise, we need to have V<N to really have less parameters to train in the model -- knowing that classical embeddings spaces size is usually between 256 and 1024, it means that we have to target a task where the number of anchors is quite low. I agree that the sparsity term on T is here to encourage to decrease the number of parameters but first, the same sparsity could be applied on the original VxN embedding matrix, and also, even if, at the end, the T matrix is sparse, during learning one has to maintain a large matrix in memory.  I would like the authors to discuss more on this point which is crucial? Particularly, I am not sure to understand what the #Emb value is in the table (AxN + AxV or just AxN), and how to compare the models. (There is a discussion in Section 3, but the argumentation does not explain why having so many parameters at train time is not a problem).  Also, since this is the crucial point in the paper, I would be interested in having a discussion about the use of neural models compression techniques after learning that could also "do the job" (even if they are not trained end-to-end).

One other remark concerns the different "components" added into the model (e.g sparsity, orthogonality, Relu...). It is difficult to measure the interest of each of them, and I would recommend the authors to provide an ablation study to make the effect of the different choices more understandable by the reader.

The notion of anchors also is misleading since it gives the impression that the A matrix will store embeddings for particular objects, while there is no constraint of that type. Each line of the A matrix is an embedding, but this embedding is not associated with one of the objects seen at train time (no direct mapping from anchors to words in the vocabulary). This has to be made more clear at the beginning of the paper.

Concerning the initialization of A by K-means, it assumes that the space of objects has a particular metric. The authors say that this metric can come from a pretrained embedding space, but in that case, the problem in the number of parameters (which is the main justification of this work) is invalid (i.e if you already have an embedding matrix, then just let us fine-tune it). Could you clarify ?

The fact that the method would allow incorporating knowledge is certainly the most interesting point. The way it is done has to be better explained (I do not understand why positive pairs are taken into account by not enforcing sparsity on T at this particular point, the way negative pairs are handled seem more natural)

The paper is interesting and proposes a new simple model that could be used to keep good performance while reducing the number of parameters of the final model. Discussions have to be added to discuss the relevance of the approach since it still needs a large number of parameters at train time, and the role of each component could be studied more in depth.

**Experience Assessment:**

I have published one or two papers in this area.

**Review Assessment: Checking Correctness Of Derivations And Theory:**

N/A

**Review Assessment: Checking Correctness Of Experiments:**

I assessed the sensibility of the experiments.

**Review Assessment: Thoroughness In Paper Reading:**

I read the paper thoroughly.

---

> ### Author Response · Authors · 2019-11-15
> **Reply to Reviewer #3 part 1**
>
> Thank you for your detailed comments and suggestions for improvements. We answer your questions and provide more experimental comparisons with baselines below.
>
> [R3 usefulness] Our methods are implemented using a dense matrix for the anchor embeddings A and a **sparse matrix** for the transformations T. Although, naively deep learning frameworks do not fully support backprop on such sparse matrix (basically change of non-zero locations in the sparse matrix is not supported) and we had do some engineering around it. In particular, for T we store only the non-zero positions and their values in a sparse format that allow efficient row slicing (adjacency list or CSOO format). The memory usage during training, storage, and evaluation are proportional to the size of A and the number of non-zero entries in T: size(A) + nnz(T). Time complexity is hard to analyse, but empirically the runtime for training does increase by 1.6 times on WikiText-103 language modeling task, but its mostly due to our unoptimized engineering. However, during inference time we see negligible difference because now native sparse ops for the T matrix can be utilized.  We do not require that V < N for our method to work, au contraire typically V >> N. In our experiments, we find that T is indeed very sparse, allowing us to obtain 10-100x compression of the embedding matrix, which in our opinion is a good trade-off. We have added these details to subsection 3.4 in the paper. We also outlined several tips to further speedup training in Appendix C and ways to incorporate our method with existing speedup techniques like softmax sampling or noise-contrastive estimation.
>
> Simply applying l1 sparsity to the entire V x N embedding matrix can be seen as a special case of our method where we use **no** anchors. This is undesirable since 1) each object is also modeled independently without information sharing between objects (from a statistical perspective, no strength in parameter sharing), and 2) there are no underlying anchors to induce the remaining representations.
>
> [R3 compression techniques] For the purposes of comparison, we selected a method based on hashing [3] as a post-processing step after training the embedding matrix. Specifically, we call Post-SH baseline where we take the trained embedding matrix from a language model trained on PTB or WikiText-103, compress the matrix using the method from [3] (k-means to obtain the anchors + sparse representation the remaining points as in Alg 1 of [3]), and use the reconstructed matrix for evaluation. As performance was not good, we tried to improve the method. In particular, we use k-SVD [4] to solve for a sparse representation instead of using ad-hoc projection methods (eq 8-9) from [3] and report it as an additional baseline which we call Post-SH+k-SVD. Comparing to these post-processing methods we demonstrate that end-to-end joint training of sparse embedding matrices is beneficial over post-processing compression.
>
> We present these results as follows:
> Using AWD-LSTM on PTB language modeling:
>
> 			        #anchors	perplexity	#params (M)
> Post-SH 		        1,000		118.8		0.60
> Post-SH		        500 		        166.8		0.30
> Post-SH+k-SVD	1,000 		78.0 		0.60
> Post-SH+k-SVD	500 		        103.5 		0.30
> ANT (ours)		1000		72.0		        0.44
> ANT (ours)		500		        74.0		        0.26
>
> Using AWD-LSTM on WikiText-103 language modeling:
>
> 			        #anchors	perplexity	#params (M)
> Post-SH 		        1,000		764.7		5.7
> Post-SH		        500 		        926.8		2.9
> Post-SH+k-SVD	1,000 		73.7 		5.7
> Post-SH+k-SVD	500 		        148.3 		2.9
> ANT (ours)		1000 		39.7 		3.1
> ANT (ours) 		500 		        54.2 		0.4
>
> We have also updated Tables 2 and 3 in the paper accordingly with these new baselines.
>
> These empirical results show that joint end-to-end training of the sparse embedding matrices is beneficial over post-processing compression, where errors may accumulate in both downstream tasks as well as embedding reconstruction. We observe that the performance improvement of ANT over post-processing compression methods is larger on WikiText-103 as compared to PTB, demonstrating that our end-to-end sparse embedding method is particularly suitable for tasks with large vocabularies. We emphasize that we are the first to incorporate these ideas of anchor points and sparse transformations into modern neural models for discrete objects.

---

> > ### Author Response · Authors · 2019-11-15
> > **Reply to Reviewer #3 part 2**
> >
> > [R3 #Emb] #Emb represents the number of non-zero embedding parameters, which is computed as size(A) + nnz(T). Our methods are implemented using a dense matrix for the anchor embeddings A and a sparse matrix for the sparse transformations T. The memory usage and time complexity during training, storage, and evaluation are therefore proportional to the size of A (since A is dense) and the number of **non-zero** entries in T (since T is very sparse). We also compute #Emb for other baselines in the same way: the total number of non-zero entries used in the embedding parameters. For methods that compress the embedding matrix into dense embedding parameters (e.g. low-rank [1], vocabulary selection [2]), this reduces to the number of parameters in the compressed form. Our method achieves the **fewest** non-zero parameters after compression (lowest #Emb) while retaining the **best** accuracy/lowest perplexity metric as compared to the baselines.
> >
> > [R3 ablations] Table 1 shows the importance of sparsity and non-negativity (relu) on T towards better compression and performance as compared to the baselines that do not use sparsity and non-negativity. We observe that sparsity is important: baseline methods that only perform lower-rank compression with dense factors (e.g. low-rank, vocabulary selection) tend to suffer in performance while using more parameters, while our method retains performance with better compression. Tables 1 and 2 (second half) also demonstrate that domain knowledge, when available, gives further boosts in performance.
> >
> > [R3 anchors] Depending on the initialization, the anchors may or may not store embeddings for particular objects. When initializing A using clustering or frequency, the anchors directly correspond to embeddings for frequent words or words at cluster centers. The case when A is initialized randomly gives anchors that do not represent any particular words as the reviewer pointed out.
> >
> > [R3 k-means] In some cases, we often start with pre-trained embedding spaces such as Glove and fine-tune for specific tasks. Instead of maintaining the entire Glove embedding matrix during training, storage, and inference, we can simply use Glove **once** to obtain initial cluster centers to initialize A. This still reduces time and space complexity during training, storage, and inference. In addition, our results show that using a dynamic (random) basis performs also well. We also note that in certain scenarios, instead of clustering across the entire vocabulary, we can take the frequent (say 20%) objects and cluster on top of those to get better coverage of the important objects. All the initialization strategies we proposed are flexible and can be optimized for specific downstream tasks.
> >
> > [R3 knowledge] T is a |V| x |A| matrix where T_{ij} represents the transformation parameter from anchor object j to object i. Ideally, we want T to be row sparse such that each object i is induced from only a few anchor objects, so we perform proximal gradient updates on T. If we know that object i is related to anchor object j (e.g. object i = canary, anchor object j = bird), then entry T_{ij} should not be sparse/equal to zero. This method implicitly takes into account both positive pairs (T_{ij} need not be sparse) and negative pairs (T_{ij} constrained to be sparse).
> >
> > References:
> > [1] Grachev et al., Compression of recurrent neural networks for efficient language modeling, arXiv 2019
> > [2] Chen et al., How large a vocabulary does text classification need? A variational approach to vocabulary selection, NAACL 2019
> > [3] Y. Guo et al., Learning to hash with optimized anchor embedding for scalable retrieval, TIP 2017.
> > [4] Aharon et al., K-SVD: An Algorithm for Designing Overcomplete Dictionaries for Sparse Representation, TSP 2006

---

### Official Review · AnonReviewer2 · 2019-10-26
**Official Blind Review #2**

**Rating:** 3

**Review:**

This manuscript proposed to represent the embedding matrix as a small set of anchor embedding and sparse transformation. The paper is trying to be general-purpose, end-to-end trainable, and able to incorporate domain knowledge. Experimental results show that it is possible to compress the embedding in the proposed way without much loss of accuracy.

The authors propose to find anchor embedding by several methods, such as frequency, clustering, or random sampling. The sparsity on the transform is imposed by L_1. Although I get the basic idea and I am familiar with many of the techniques, it is unclear to me what is the main focus of this paper, and the technical contribution is quite vague. Why is the large embedding matrix a problem? Besides the low-rank form proposed, are there any other ways to compress it? This paper is not well motivated at all. Therefore, I think this manuscript is not ready to publish in its current form.

#####
Thank you for the response! I've increased my score to 3: Weak Reject. Although the idea of compressing the (word) embedding layer using low-rank structures is not new (even with the end-to-end training), the main technical contribution in this paper is to jointly learn the anchor embedding (anchor pre-selected with multiple schemes) and sparse transformation (sparsity achieved via Proximal GD). Moreover, domain knowledge can be incorporated by adding specialized constraints such as orthogonality and selective penalization.

At first glance, the idea presented in this paper seems not new, and I doubt many people are doing similar stuffs already in practice. I find the explanations on the technical points in Appendix C helpful. The empirical study in this paper looks strong. The authors considered experiments in text classification and language modeling with a number of baselines, which demonstrates the advantages of anchor and joint training in the proposed way. This paper presents several useful heuristics around, but I share the concern with other reviewers about whether the main point is compelling enough, given the existing body of work along with this line.







**Experience Assessment:**

I do not know much about this area.

**Review Assessment: Checking Correctness Of Derivations And Theory:**

I assessed the sensibility of the derivations and theory.

**Review Assessment: Checking Correctness Of Experiments:**

I assessed the sensibility of the experiments.

**Review Assessment: Thoroughness In Paper Reading:**

I made a quick assessment of this paper.

---

> ### Author Response · Authors · 2019-11-15
> **Reply to Reviewer #2**
>
> Thank you for your detailed comments and suggestions for improvements. We answer your questions below.
>
> [R2 embedding matrix] Why is the large embedding matrix a problem?
>
> When training neural models for text, documents, URLs, users, queries, or any other problem involving large sets of discrete objects, the embedding matrix scales prohibitively with the number of objects and takes up most of the parameters across the entire model. It is therefore desirable to reduce the size of the embedding matrix for more efficient learning, storage, and inference. The reviews from both of the other reviewers, as well as the long line of related work in this area (section 2 of our paper), indicates that there is much interest in this area and train models with much fewer parameters that work as well as large models.
>
> [R2 other methods] Besides the low-rank form proposed, are there any other ways to compress it?
>
> Section 2 of our paper describes several lines of related work on compressing the embedding matrix. In summary, these baseline methods span methods based on low-rank approximations, quantization, hashing, vocabulary selection, and codebook learning. We also compared our proposed method with these baselines in our experiments (Tables 1, 2, 3), showing that our method outperforms the existing compression baselines on text classification and language modeling tasks. We would like to emphasize that we are the first to incorporate these ideas of anchor points and sparse transformations into modern neural models for discrete objects, demonstrating strong performance on several tasks involving representation learning of discrete objects.

---

### Official Review · AnonReviewer1 · 2019-11-08
**Official Blind Review #1**

**Rating:** 3

**Review:**

This paper proposes a general embedding method, Anchor & Transform (ANT), that learns sparse representations of discrete objects by jointly learning a small set of anchor embeddings and a sparse transformation from anchor objects to all the others.

Strengths of the paper:
1. The paper is well-written and easy to be followed.
2. The research problem is of great value to be investigated.

Weaknesses of the paper:
1. The idea of utilizing anchors to reduce the size of features (in your case, the total embeddings of discrete objects to be inferred) has been widely studied in related fields in computer science. For instance, there are a number of papers in the field of manifold learning using anchors to reduce the size. The inherent connections and relationships between the proposed methods and other algorithms using anchors should be carefully discussed.
2. The contribution of the paper seems not significant, as the idea of utilizing anchors to reduce the number of parameters to be inferred has been widely studies in the related work. There are a number of papers utilizing anchors, such as the followings (just list some of them):
B. Xu et al., Efficient manifold ranking for image retrieval, in SIGIR 2011.
S. Liang et al., Manifold learning for rank aggregation, in WWW 2018.
Y. Guo et al., Learning to hash with optimized anchor embedding for scalable retrieval, in TIP, 2017.
In the last reference aforementioned above, both anchors and embeddings are jointly taken into account.
3. Baselines should include some manifold learning algorithms that take anchors into account.


**Experience Assessment:**

I have published one or two papers in this area.

**Review Assessment: Checking Correctness Of Derivations And Theory:**

I carefully checked the derivations and theory.

**Review Assessment: Checking Correctness Of Experiments:**

I carefully checked the experiments.

**Review Assessment: Thoroughness In Paper Reading:**

I read the paper thoroughly.

---

> ### Author Response · Authors · 2019-11-15
> **Reply to Reviewer #1**
>
> Thank you for your detailed comments and suggestions for improvements. We answer your questions and provide more experimental comparisons with baselines below.
>
> [R1 related work and contributions] While the works you cited do indeed use the concept of ‘anchors’ to represent a space of objects, our main contribution was to demonstrate how these anchors and sparse transformations can be **trained jointly** with neural models as a general input embedding layer, and how we can obtain better sparse representations using domain knowledge. The methods in papers listed by you apply when we have access to some similarity or distance function between the objects (or its proxy like ranked examples).  One can definitely apply those methods as a post-processing step to reduce features while preserving the embedding space more or less (More on this in next bullet). We have added a paragraph and cited some of the works in this area. However, it is not clear how to apply those methods with arbitrary functions a deep net classifier or language model is learning. We are the first to present this general approach and demonstrate its effectiveness of a suite of tasks involving representation learning of discrete objects.
>
> [R1 baseline comparisons] We implemented the method based on hashing [1] as a post-processing step after training the embedding matrix. Specifically, we call Post-SH baseline where we take the trained embedding matrix from a language model trained on PTB or WikiText-103, compress the matrix using the method from [1] (k-means to obtain the anchors + sparse representation the remaining points as in Alg 1 of [1]), and use the reconstructed matrix for evaluation. The performance was not very good, so we tried to improve the method. In particular, we use k-SVD [2] to solve for a sparse representation instead of using ad-hoc projection methods (eq 8-9) from [1] and report it as an additional baseline which we call Post-SH+k-SVD. Comparing to these post-processing methods we demonstrate that end-to-end joint training of sparse embedding matrices is beneficial over post-processing compression.
>
> We present these results as follows:
> Using AWD-LSTM on PTB language modeling:
>
> 			        #anchors	perplexity	#params (M)
> Post-SH 		        1,000		118.8		0.60
> Post-SH		        500 		        166.8		0.30
> Post-SH+k-SVD	1,000 		78.0 		0.60
> Post-SH+k-SVD	500 		        103.5 		0.30
> ANT (ours)		1000		72.0		        0.44
> ANT (ours)		500		        74.0		        0.26
>
> Using AWD-LSTM on WikiText-103 language modeling:
>
> 			        #anchors	perplexity	#params (M)
> Post-SH 		        1,000		764.7		5.7
> Post-SH		        500 		        926.8		2.9
> Post-SH+k-SVD	1,000 		73.7 		5.7
> Post-SH+k-SVD	500 		        148.3 		2.9
> ANT (ours)		1,000 		39.7 		3.1
> ANT (ours) 		500 		        54.2 		0.4
>
> We have also updated Tables 2 and 3 in the paper accordingly with these new baselines.
>
> These empirical results show that joint end-to-end training of the sparse embedding matrices is beneficial over post-processing compression, where errors may accumulate in both downstream tasks as well as embedding reconstruction. We observe that the performance improvement of ANT over post-processing compression methods is larger on WikiText-103 as compared to PTB, demonstrating that our end-to-end sparse embedding method is **particularly suitable** for tasks with large vocabularies. We would like to emphasize that we are the first to incorporate these ideas of anchor points and sparse transformations into modern neural models for discrete objects.
>
> References:
> [1] Y. Guo et al., Learning to hash with optimized anchor embedding for scalable retrieval, TIP, 2017.
> [2] Aharon et al., K-SVD: An Algorithm for Designing Overcomplete Dictionaries for Sparse Representation, TSP 2006

---

### Author Response · Authors · 2019-11-15
**Submission Update**

We thank all the reviewers for their detailed comments and suggestions for improvements. We have incorporated your feedback and updated the paper accordingly. Here we outline the main additions to the paper:

Tables 2 and 3: we added comparisons with 2 post-processing compression methods based on clustering and sparse hashing [1], as well as further improving this using sparse coding with k-SVD [2]. Table 2 shows results for PTB language modeling and Table 3 shows results for WikiText-103 language modeling. We outperform both methods and we hypothesize this is because post-processing compression accumulates errors in both language modeling as well as the embedding reconstruction. We observe that the performance improvement of ANT over post-processing compression methods is larger on WikiText-103 as compared to PTB, demonstrating that our end-to-end sparse embedding method is particularly suitable for tasks with large vocabularies. We conjecture that as vocabulary size increases, running clustering becomes harder, e.g. good initializations like KMeans++ become prohibitively expensive. We describe implementation and hyperparameter details for these 2 post-processing baselines in appendix E2. We also improve compression for WikiText-103 language modeling task for ANT by a better hyper-parameter search. New result is updated in Table 3.

Subsection 3.4: we have added details regarding the time and memory complexity of training our sparse embedding layer. We implement our method using a dense matrix for the anchor embeddings A and a **sparse matrix** for the sparse transformations T. Although, naively deep learning frameworks do not fully support backprop on such sparse matrix (basically change of non-zero locations in the sparse matrix is not supported ) and we had do some engineering around it. In particular, for T we store only the non-zero positions and their values in a sparse format that allow efficient row slicing (adjacency list or CSOO format). The memory usage during training, storage, and evaluation are proportional to the size of A and the number of non-zero entries in T: size(A) + nnz(T). Time complexity is hard to analyse empirically, but the runtime for training does increase by 1.6 times on WikiText-103 language modeling task, but its mostly due to our unoptimized engineering. However, during inference time we see negligible difference because now native sparse ops for the T matrix can be utilized. In our experiments, we find that T is indeed very sparse, allowing us to obtain 10-100x compression of the embedding matrix, which in our opinion is a good trade-off. We also outlined several tips to further speedup training in Appendix C and ways to incorporate our method with existing speedup techniques like softmax sampling or noise-contrastive estimation.


[1] Y. Guo et al., Learning to hash with optimized anchor embedding for scalable retrieval, TIP, 2017.
[2] Aharon et al., K-SVD: An Algorithm for Designing Overcomplete Dictionaries for Sparse Representation, TSP 2006

---

### Decision · Program_Chairs · 2019-12-19

**Decision:**

Reject

**Comment:**

The paper proposes a method to produce embeddings of discrete objects, jointly learning a small set of anchor embeddings and a sparse transformation from anchor objects to all the others. While the paper is well written, and proposes an interesting solution, the contribution seems rather incremental (as noted by several reviewers), considering the existing literature in the area.  Also, after discussions the usefulness of the method remains a bit unclear - it seems some engineering (related to sparse operations) is still required to validate the viability of the approach.